# Implementing novel, flexible, and powerful survey designs in R Shiny

**Aaron R. Kaufman** *

Division of Social Science, New York University Abu Dhabi, Abu Dhabi, United Arab Emirates

* aaronkaufman@nyu.edu

## Abstract

Survey research is ubiquitous in the social sciences as a cost-effective and time-efficient means of collecting data. However, the available software for implementing and disseminating such surveys lacks flexibility, stifling researcher creativity and severely limiting the scope of questions that survey research can address. In this paper I introduce the use of R Shiny, an open source web application and scripting language, for implementing powerful, innovative, and fully customizable surveys. Through six applications rooted in important questions in political science, I show that R Shiny allows for (1) randomized question selection, (2) programmatic treatments, (3) programmatic survey flow, (4) adaptive question batteries, (5) sequentially block-randomized design, and (6) randomized intracoder reliability tests, expanding the scope, ease, and cost effectiveness of online survey research. I make all replication code available online.

## 1 New research methods open new questions

New innovations in research methodology historically precede bursts in research productivity. Bigger and more advanced telescopes engender new theories of the universe; gene editing technologies prompt the development of hardier, more nutritious fruits and vegetables; powerful computational systems facilitate ingesting huge quantities of data, synthesizing information in new ways to understand human interactions at scale. The most significant advances, rather than merely answering existing questions, open new lines of inquiry never before conceived.

Among the most significant advances in political science was the advent of survey research, which allowed researchers to shift from considering only aggregations of actors to rigorously interrogating how *samples* of individuals conceive of their political contexts and how those notions affect their behaviors. Survey research has subsequently improved through innovations like the Computer-Assisted Telephone Interview [1] which enabled researchers to automatically survey random samples of households, and later, online survey software allowing researchers to conduct more advanced randomization and to implement advanced survey flow. More recently, services like Amazon.com's Mechanical Turk (MTurk) and the Digital Laboratory for the Social Sciences (DLABSS) provide low-cost or free survey respondents, rendering this rich data source and research methodology widely accessible [2].

**Data Availability Statement:** All software described in this paper is available at https://github.com/aaronrkaufman/shinySurveys.

**Funding:** The author(s) received no specific funding for this work.

**Competing interests:** The authors have declared that no competing interests exist.

Today, surveys and survey experiments are among the most valuable sources of data across psychology [3], sociology [4], political science [5], and behavioral economics [6]. Survey analysis is a rich literature in statistics [7], and survey design includes decades of research in the social sciences [8–10].

Despite its ubiquity, current survey software limits the scope of questions that survey researchers might address. Important lines of inquiry, even those well-supported by current statistical techniques, simply cannot be pursued, due either to technical concerns such as rigid and inefficient proprietary software or to practical concerns like sample size, statistical power, or researcher time and money.

I introduce a technological upgrade to online survey research to increase its scope of design, statistical efficiency, cost-effectiveness, and flexibility: building custom surveys using R Shiny, an extension of the popular R programming language. The software built in R Shiny could be built in any high-level language like JavaScript or Ruby; R Shiny has a singular advantage in social science research in that graduate students learn R by default. Writing or modifying R Shiny code is much more feasible given the pre-existing skill set of quantitative social scientists. As well, a diverse ecosystem of third-party software libraries exist for R Shiny to enhance and streamline its capabilities. I show, across six applications addressing important debates in political science, how this upgrade enables us to ask certain questions for the first time and answer old questions better. Accompanying this paper is easily adaptable open-source code to implement all six examples at https://github.com/aaronrkaufman/shinySurveys.

## 1.1 Key advantages of R Shiny

R Shiny's value as a survey research tool arises from its power and flexibility compared to standard proprietary tools (see Section 2.7). Much springs from this flexibility, but three advantages in particular empower R Shiny as a marked improvement in online survey technology. I describe these advantages in the abstract below, then illustrate their value in Section 2.

**Integrating external data.** Surveys are typically self-contained entities; if researchers need respondents to react to outside information, they must either program that information directly into the survey or else query external databases using complicated JavaScript. R Shiny allows researchers to pre-load data into the survey instrument such that when a respondent answers a question, the survey can *automatically* link that response to external information.

**Programmatic questions.** In designing an online survey instrument, researchers (1) write survey questions and (2) program survey flow. R Shiny allows researchers to program *both* survey questions and survey flow with unconstrained complexity. Survey flow can be a stochastic rather than deterministic process, for example, and survey questions can be constructed from data rather than laboriously entered into a proprietary user interface.

**Custom interaction modes.** Most survey questions fall into one of very few categories: multiple choice, free response, sliders, rank ordering. These question types are the ones enabled by standard survey tools, while more complicated survey tasks such as the Implicit Attitudes Test are usually done outside the survey context. R Shiny enables a limitless set of possible survey tasks including drag-and-drop, shape drawing, interactive games and puzzles, and many other ways to collect respondent data.

## 2 Six applications

To illustrate the advantage that R Shiny offers as a survey implementation platform, I introduce six substantively-motivated questions with research designs beyond the reach of conventional survey software; accompanying this paper is a code repository and replication archive to implement the algorithms described herein. These research questions address a broad cross-

section of behavioral research in political science: measuring media bias, estimating the efficacy of the interest group endorsement heuristic, measuring legislative district compactness, estimating audience costs. Two additional examples offer methodological insights, the first an improvement on measuring latent traits from surveys and the second an analysis of the conjoint experimental design. The first three of these six examples (heuristics, media bias, and latent trait batteries) are merely very challenging or resource-intensive to implement without R Shiny; the final three (audience costs, legislative district compactness, and conjoint design) are impossible using only current survey software, requiring advanced programming and communications with a database outside the survey platform. I motivate each design with a research question embedded in an important academic literature, then briefly describe how R Shiny enables these designs.

## 2.1 Testing the interest group endorsement heuristic with respondent-adaptive questions

Do interest group endorsements inform voters about the issue positions or roll call votes of their Members of Congress? If, for example, a voter is informed that the League of Conservation Voters gave their Member of Congress an A+ rating, is that voter better able to guess how their representative voted on the Keystone XL oil pipeline? How does this informational cue perform compared to the party identification heuristic? Broockman et al [11] interrogate this question using a series of nine experiments across four original survey samples, finding that a positive endorsement, regardless of its source, increases respondents' approval of their representative, and increases their perception of ideological similarity with that representative.

In the control condition of the first such study, respondents receive questions of the following form: "Did [respondent's representative in the US House] vote for [policy text]?" In the first treatment condition, respondents also see an informational cue: "Before you answer this question, here's some information you might find relevant: Representative [Rep.] is a member of the [respondent's representative's party] Party." The second treatment condition relates to interest group endorsements: "Before you answer this question, here's some information you might find relevant: Representative [Rep.] received a score of [x]% from the [Interest Group]." To implement this survey, the survey first asks respondents to provide their nine-digit ZIP code and uses that to identify each respondent's representative in the U.S. House of Representatives, that representative's party affiliation, and various interest groups' endorsement ratings. Since the text of the informational cue depends on respondents' answers to the nine-digit ZIP code question, implementing this survey requires *response-dependent question selection*.

There are at least two ways to implement this survey on a conventional survey platform. The first involves manually programming every nine-digit ZIP code in the US along with the Member of Congress and possible interest group ratings that ZIP code entails; this is not a practical solution. The second solution requires knowledge of JavaScript to query a web API and pipe the results from that API query into the text of the treatment condition. Few social scientists are trained in JavaScript, and even so, with this solution respondents may face latency if the web query is slow.

The R Shiny solution is parsimonious and quick, deriving from R Shiny's facility in integrating external data sources (see Section 3.2.2). Rather than query an API, after respondents indicate their nine-digit ZIP code, the survey software locates that entry on a data frame containing ZIP codes linked to Members of Congress, their party, and their interest group ratings, a fast subsetting task requiring a single line of R code.

## 2.2 Measuring media bias with thousands of possible survey questions

Researchers have identified two primary components to news media bias [12]. *Presentation bias*, also called "spin" or "slant," refers to language and the framing an article uses to discuss an issue or event [13]. *Topic selection bias*, on the other hand, refers to what topics are covered at all [14]. A conservative news outlet might cover a story in a conservative way while a liberal outlet covers that same story with a liberal bias. On the other hand, both outlets *might* cover the same story similarly, but the liberal outlet might choose to cover events that paint liberals in a positive light while conservatives might ignore those events altogether. Disentangling the relative effects of presentation bias versus selection bias is difficult.

Mozer et al [15] isolate these component effects and control for topic selection by comparing the language used among news articles covering *the same story*. Rather than identify pairs of articles covering the same story by hand, this study develops an automated method to measure news article similarity. Many methods exist for measuring the similarity of two text documents: cosine similarity, Jaccard distance, Mahalanobis distance, word embedding distance, topic loading distance. But which of these methods best captures whether two news articles cover the same story? To resolve this empirical question, the authors conduct a survey experiment. Survey respondents receive a random pair of newspaper articles and are asked to rate the two articles' content from least to most similar according to a rubric. Each pair shown to a respondent is drawn randomly from a set of 20,000 possible pairs. The technological challenge in this survey is programming all 20,000 pairs that might be presented to respondents.

As in the example above, there are two solutions under the constraints of commonly-used software. The first is to manually type all 20,000 pairs of news articles into the survey software user interface and then randomly sample from that list of 20,000 pairs. This is an incredibly time-intensive process. The alternative is to build, populate, and host a SQL server or API containing the 20,000 pairs, and then write JavaScript code to query that SQL server or API. This is a substantially larger task than simply querying a pre-existing API as in Section 2.1 and one with a higher potential for web-based latency.

The R Shiny solution combines an integrated external data source (as in Section 2.1) with a programmatic question selection algorithm (Section 3.2.1) to create a R data frame with the 20,000 pairs of news articles and `sample()` from that data frame. This is trivially accomplished in two lines of beginner R: the first to load the data frame, and the second to sample a row from it.

## 2.3 Capturing latent traits with survey batteries

Latent trait batteries are crucial in political science, psychology, and other behavioral disciplines for capturing complex, nuanced, and multifaceted features of respondents' personalities. Such batteries aim to elicit theoretically important traits like the Big Five personality traits [16, 17], narcissism [18], political knowledge [19–21], prosocial inclination [22], social dominance orientation [23, 24], and symbolic racism [25, 26], among many others.

Despite these batteries' important role in survey-based measurement, they are imperfect. The full batteries for many of these concepts last for 40 questions or more, contributing to respondent fatigue and measurement error. Along the way they often collect redundant information. Consider a political sophistication battery: if a respondent correctly answers a very difficult question indicating a high political knowledge, asking a much easier question is inefficient, as that respondent is very likely to answer that question correctly. To reduce the risk of respondent fatigue and to mitigate redundant information, Montgomery and Rossiter [27] show that it is optimal to ask these batteries *adaptively*, taking cues from the Graduate Record Examinations (GRE). This is an example of *response-dependent question selection*: a

respondent's answer to the first question determines the second question, and their response to that second question determines the third question.

This results in an exponentially-expanding survey programming task. Since each answer to every question entails a different subsequent question, writing an 8-question adaptive battery where each question only has two answers involves programming 255 logical conditions into the battery alone; if each question has more than two answers, that number of logical conditions grows rapidly. While this is possible to do in software like Qualtrics, it is severely time-prohibitive.

Montgomery and Rossiter's adaptive latent trait model, an illustration of programmatic questioning, is implemented as a R library (Section 3.2.2). Implementing it requires three lines of R code: the first to initialize the latent trait battery, the second to update the battery after each response, and a third to return the next question. An overview of this algorithm is in Section 3.2.2.

## 2.4 Detecting audience costs with sequential pair randomization

An important literature in international relations asks: Do voters punish politicians for appearing to back down after escalating an international dispute [28, 29]? Audience costs refer to hypothesized penalties, electoral and otherwise, that leaders pay when they appear inconsistent or irresolute in foreign conflicts by publicly threatening to take military action and subsequently reneging. In a canonical survey experiment, Tomz [30] tests the existence and strength of audience costs using a randomized vignette experiment, and finds significant support for the audience costs hypothesis across a broad cross-section of the population and across diverse contexts, largely driven by citizens' concerns about their leader's and country's international reputation.

In this experiment, survey takers read a vignette about a hypothetical international conflict and respond by indicating their (dis)approval of their leader's actions. The control vignette reads: "The U.S. president said the United States would stay out of the conflict. The attacking country continued to invade. In the end, the U.S. president did not send troops, and the attacking country took over its neighbor." The treatment vignette illustrates backing down after escalating: "The U.S. president said that if the attack continued, the U.S. military would push out the invaders. The attacking country continued to invade. In the end, the U.S. president did not send troops, and the attacking country took over its neighbor."

In order to prevent idiosyncratic features of any singular crisis from driving the results, the study randomly varies four contextual variables: regime, motive, power, and interests. The attacking country was led by a "dictator" or "democratically elected government." The attacker had aggressive motives "to get more power and resources" or less aggressive motives of "a long-standing historical feud." The attacker had a "strong military" such that "it would have taken a major effort for the United States to help push them out" or a "weak military" that the United States could repel without major effort. Finally, victory by the attacking country would "hurt" or "not affect" the safety and economy of the United States.

These four contextual factor variables are strongly predictive of the outcome. To improve the precision with which this experiment estimates audience cost treatment effects, one might sequentially pair-randomize on these contextual factor variables, reducing the necessary sample size while improving precision. Sequential blocking on covariates gathered before treatment (as is the case in Tomz [30]) can improve precision in experimental analysis, reducing standard errors by as much as a factor of 6 [31]. This precision gain is valuable for achieving statistical significance in low-powered experiments or subgroup analyses, but it also opens

new research questions in domains where effect sizes are typically small but are strongly correlated with pre-treatment covariates.

**2.4.1 Sequential pair-randomization.** In sequential pair-randomization, respondents are randomized based on their covariate profile. The first time a covariate profile appears, the sequential pair-randomization procedure flips a coin and assigns that respondent to treatment or control. The next time that covariate profile appears, the procedure assigns the *opposite condition* as the first time. The third time, it flips a coin again, and so on.

Implementing this procedure in standard survey software is an incremental step more difficult even than that of Section 2.4. The first component is a SQL server or API to store respondent covariate profiles and their treatment assignments as they arrive to take the survey. The second component is a snippet of JavaScript code to query the most recent treatment assignment for a new respondent's covariate profile, determine that new respondent's treatment assignment, and then update the SQL server or API back-end with the new respondent's data.

The R Shiny solution involves two components: an data frame with respondent information loaded into R using the `load()` function and the R library `blockTools` [32], which automatically produces a treatment assignment from that stored data frame. This external data frame might be stored on an R Shiny server to avoid latency, or in an external location like Dropbox using the `rdrop2` library. For a more detailed explanation, see Section 3.2.4.

## 2.5 Measuring legislative district compactness with dynamic drag-and-drop tools

The U.S. Constitution requires that states redraw their Congressional districts every decade, following the Census, to reapportion seats in the U.S. House of Representatives. This contentious process follows a set of traditional redistricting principles which can deter or surmount claims of race-based gerrymandering; among the most ambiguous of these is *compactness*. This word has an intuitive definition related to a shape being, according to the Oxford English Dictionary ("close or neatly packed together") but it also has a series of formal, quantitative metrics produced by political scientists [33, 34] that frequently contradict each other in their relative assessments of districts.

Kaufman et al [35] address this important inconsistency by asking respondents to rank-order legislative districts according to their perceived compactness using a bubble-sort algorithm and training a supervised machine learning model to capture the relationship between respondent perceptions and several dozen pre-existing compactness measures. Then, to validate this supervised learning method, the authors implement a validation study where many respondents rank-order a set of legislative districts using a drag-and-drop interface.

In the first survey, respondents are assigned a random sample of six legislative districts, shown two at a time, and asked to identify which item in each pair is more compact according to their own judgment. R Shiny's bubble-sort can produce a sorted list of legislative districts with as few as six comparisons rather than the 30 it would require to estimate every choice without a bubble-sort algorithm. This bubble sort is easily implemented in R (see Section 3.2.3). The second survey uses a rank-ordered drag-and-drop procedure, an example of a custom interaction mode, in which respondents move district images around on their screen to produce an ordered list (see Fig 1). The authors then use respondent-derived district rankings to build a machine learning model of legislative district compactness and show that their measure correlates strongly with federal judges' preferences for compact districts.

# Legislative District Compactness Study

The law requires that legislative districts for the US congress and many state legislatures be "compact". The law does not say exactly what district compactness is, but generally, people think they know it when they see it. One dictionary definition of compactness is "joined or packed together closely and firmly united; dense; arranged efficiently within a relatively small space."

Here's your task: Below is a group of legislative districts, randomly ordered. Order the districts from most compact (at the top left) to least compact (at the bottom right) according to your own best judgement, by dragging and dropping. We have many individuals performing this task, and the more your ranks are similar to others like yourself, the better you will have done.

**MOST Compact Here**

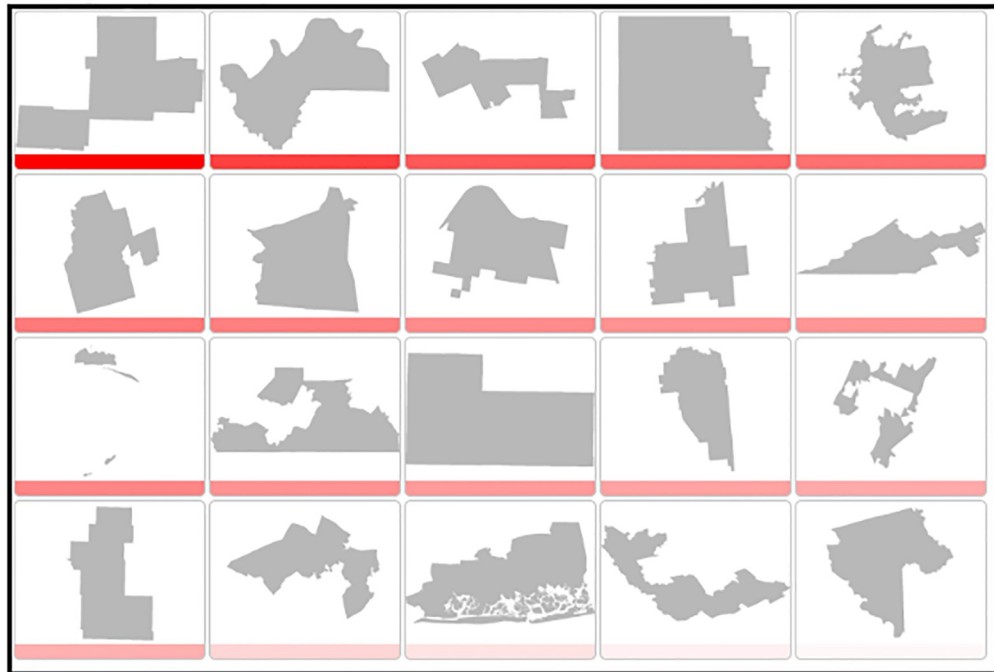

**LEAST Compact Here**

**Fig 1. R Shiny interface.** An image of the drag-and-drop R Shiny interface for obtaining a rank ordering of legislative district compactness; this interaction mode is not possible in standard survey software.

## 2.6 Measuring the measurement error of conjoint tasks

Conjoint designs, originated in marketing research [36, 37], are increasingly popular in political science [38–40]. Such designs involve presenting respondents with fully-randomized pairs of profiles in sequence, and for each pair, asking the respondents to select which one they

prefer. Such profile pairs might include political candidate resumés, newspaper articles, potential immigrant dossiers, or proposed building development specifications. Conjoint designs are preferable for such comparisons in part because they produce stable estimates of many difficult-to-isolate effects simultaneously and show "which components of a multidimensional treatment are influential" [38]. Yet recent work casts doubt on the validity of conjoint analysis in important contexts [41].

Clayton et al. [42] study one threat to this validity in particular: intracoder reliability. Analysis of a conjoint experiment assumes that when a respondent selects one profile in a pair, that choice is definitive; if that choice is probabilistic, subsequent results may be misstated. To interrogate this threat, the authors undertake an intracoder reliability study. Each respondent receives ten (unique) fully-randomized conjoint tables and selects their preferred profile for each; two weeks later, each respondent receives the same ten tables as before in a different order and again selects their preference.

A Qualtrics implementation of this requires, as in the above examples, an external SQL database with a mapping between respondent identifiers and a set of conjoint tables as well as a JavaScript snippet to query that database. The R Shiny solution leverages an integrated external data source by setting R's internal random seed according to each respondent's unique identifier then generating the randomized conjoint tables to ensure that a respondent receives the same questions in wave 2 as in wave 1.

In doing so, they find that the average intercoder reliability of conjoint studies is about 75%—higher than many other survey measures, but with substantial implications for estimating causal effects.

## 2.7 Further benefits and considerations

Survey research is a demanding enterprise and attention to detail is vital in survey design and implementation, yet many survey researchers and industry practitioners relinquish control over many of these important details to the default settings of software like Qualtrics. R Shiny gives survey designers the flexibility to control every facet of their survey's appearance and function; it opens up new possibilities in programmatic survey design, integrating external data, and novel interaction modes, and the applications above are far from exhaustive. But it comes with additional features inherent in R systems, as well, to the great benefit of survey research. The first two key advantages contribute to open and reproducible science, while the next four further expand the scope and quality of research. For a full side-by-side comparison of R Shiny and Qualtrics, including visual comparisons, see Section 3.1.

1. R Shiny applications can be easily open-sourced on Github or Dataverse for research design transparency

2. R Shiny applications can perform randomization with a pre-specified random seed, allowing easy reproducibility

3. R Shiny applications are accessed through a URL which can receive optional arguments like Mechanical Turk IDs or link IDs to identify respondents

4. R Shiny applications have a wide variety of survey interaction modes such drag-and-drop functionality that commercial software does not

5. R Shiny itself is open-source, and an active community of developers produce new features regularly

6. The R Shiny user interface is fully customizeable, stripping away unnecessary aesthetic elements.

## 3 Discussion and new lines of inquiry

Scholars tend to design research around the tools they have for implementing that research. This paper introduces R Shiny as a tool to implement complex survey designs and randomization schemes, opening new avenues of research where current survey software proves too rigid or cumbersome, and expanding the scope of survey research available to academics and practitioners. This R Shiny framework allows some survey designs that are impossible in currently available software, such as sequential blocked randomization, and others that are merely infeasible, such as programmatic survey flow. I have presented six cases demonstrating how R Shiny can be used to answer new questions for the first time, or old questions with new facility. Finally, I have open-sourced the R Shiny application code for all six examples, allowing future researchers to develop new survey applications in line with those illustrated here or in new directions entirely.

The applications I detail in Section 2 are far from exhaustive. I offer three areas where R Shiny may offer practical improvements in the future. First, adaptive experiments are common practice in the health sciences—as more observations are collected and treatment effect estimates become more precise, experimenters may wish to allocate more respondents to the treatment condition to improve their health outcomes—and implementing similar methods in R Shiny may be useful for designing optimally informative or persuasive treatments. Secondly, a growing literature on conjoint designs may soon develop best practices for reducing measurement error; an adaptive survey instrument that, for example, reduces the experiment's complexity if measurement error appears too high would enormously improve the results from conjoint studies. Finally, by allowing researchers to link survey responses to multiple external data sources *mid-survey*, R Shiny may enable researchers to tailor surveys to respondents in new ways. Consider, for example, a two-wave panel where the second-wave survey first links to a respondent's Wave 1 answers, and can ask respondents about changes from Wave 1 to Wave 2. Such a survey design could offer rare insight into the mechanisms for opinion change.

Online survey research has been a tremendous boon to research in the social sciences, reducing barriers to entry in experimental studies and hastening the pace of behavioral research. The tools I have introduced in this paper build on existing online survey methods and both increase the scope of potential online survey designs and reduce field work costs associated with studies, expanding the set of questions researchers can address with survey research.

### 3.1 R Shiny

R Shiny was designed for, and is primarily used as, an interface for interactive data visualization. An R Shiny application involves two main functions: a *server* function ("server.R") and a *user interface* (UI) function ('ui.R'). The UI function, modifiable with Cascading Style Sheets (CSS) and Hypertext Markup Language (HTML), builds the user-facing side of the application, while the server function performs the calculations in R. Like other R programming, R Shiny can load libraries, define functions, perform operations, produce output files or plots, and interact with the internet. For example, the *UI* function might build a drop-down menu with a default and several additional options as *Input 1*. The server function includes a function to

**Table 1. Comparison of key features, Qualtrics versus R Shiny.** R Shiny offers greater flexibility on all dimensions than the proprietary industry standard Qualtrics.

|  | Qualtrics | R Shiny |
|---|---|---|
| Coding the survey | Entirely manual | Manual or programmatic |
| Survey Flow | Deterministic, pre-specified | Dynamic, adaptive |
| Sharing | Requires paid Qualtrics account, or export as PDF | Open-source, GitHub |
| Randomization | Opaque | Flexible, can set seed |
| Pass arguments through links | No | Yes |
| Question types | Multiple Choice, Rank order, sliders, text entry | Fully customizeable |
| Source code available | Fully proprietary | Fully open-source |
| In active development | Yes, slowly | Yes, rapidly |
| Interface | Pick from themes, edit CSS | Fully customizeable |
| Desktop versus mobile | Often appear different | Always appear the same |

recognize when *Input 1* changes, and then runs standard R code to perhaps output a plot. This plot might render in a separate section of the user interface.

This interface can change dynamically, and the server can perform arbitrarily complex R code, giving R Shiny enormous potential for user-facing applications. In this paper I focus on the use of R Shiny as a survey platform. Not discussed in this paper is its use as an API: researchers can build interfaces to complex statistical or machine learning models trained on proprietary data sets, allowing users access to the model without compromising data or entailing significant start-up costs.

Excellent tutorials exist for R Shiny, such as https://shiny.rstudio.com/tutorial/.

Table 1 below outlines and explains the differences between R Shiny and the industry-standard alternative, Qualtrics, for online survey research. Figs 2, 3 and 4 show some of the aesthetic differences between R Shiny and Qualtrics, including an example of how Qualtrics performs poorly on mobile devices (R Shiny is always identical on mobile as on a desktop).

**Comparison 1 of 13**

Below are several pieces of information about hypothetical housing developments that might be built in your city or town. Please indicate which of the two housing developments you would personally prefer to be built in your city or town.

|  | Building 1 | Building 2 |
|---|---|---|
| How many units will the building have? | 12 units | 48 units |
| How is the land currently used? This will be demolished. | Historically-designated building | Parking lot |
| How far is the building from your home? | 1/2 mile (10 minute walk) | 1/8 mile (2 minute walk) |
| How tall will the building be? | 6 stories | 12 stories |
| How do local residents feel about the building? | No opinion | Oppose the building |
| What share of units will be affordable to low-income residents? | None of the units | Half of the units |
| Will residents own or rent? | Rent | Own |

[ Building 1 ] [ Building 2 ]

[ Next ]

**Fig 2. Conjoint in R Shiny.** An example conjoint study in R Shiny.

Which candidate do you prefer?

| | Candidate A | Candidate B |
|---|---|---|
| Partisanship | Democrat | Republican |
| Gender | Male | Female |
| Imposing restrictions on abortion rights | Support | Support |
| Allow gay couples to be legally married | Support | Oppose |
| A carbon tax ("cap and trade") system for greenhouse gases | Support | Oppose |
| Increased taxes on those making over $250,000 | Support | Support |
| Increased spending on national defense | Oppose | Support |

| Candidate A | Candidate B |
|---|---|

**Fig 3. Conjoint in Qualtrics.** An example conjoint study in Qualtrics.

### 3.2 Algorithms

This section details the flow of R Shiny survey algorithms designed for the applied research problems in Section 2.

#### 3.2.1 Random question selection algorithm.

1. Respondents are directed to the online survey.

2. R Shiny Server selects a random subset of questions.

3. The respondent receives the random set of questions, and responds.

4. An outcome file is stored locally, or uploaded to an external repository for analysis.

5. If applicable: the respondent receives a random hash code to receive compensation.

An implementation of this algorithm, applied to the study of media bias, is found here: https://github.com/aaronrkaufman/shinySurveys/tree/master/media_bias.

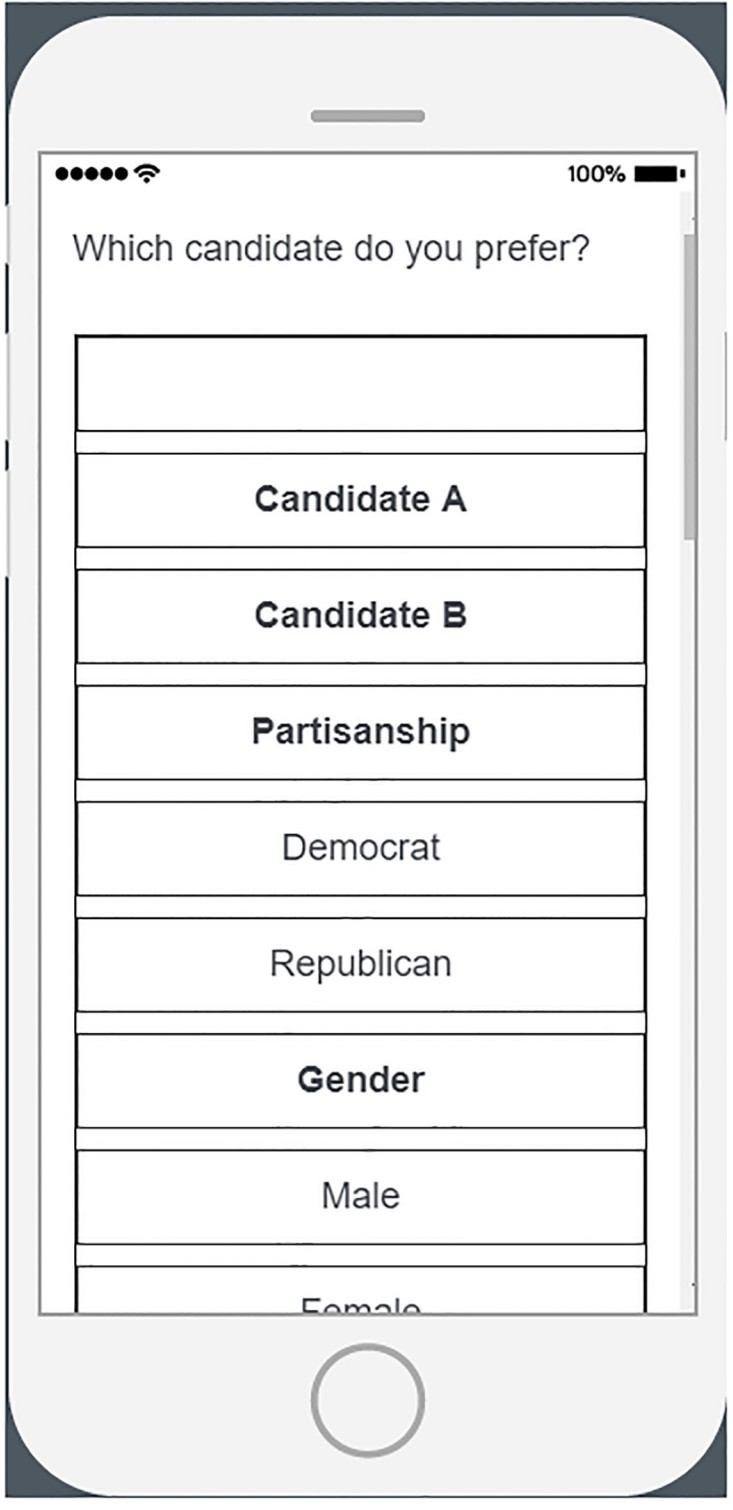

**Fig 4. Conjoint on Qualtrics mobile.** An example conjoint study in Qualtrics on a mobile device. Standard HTML tables do not render well on mobile devices in Qualtrics.

### 3.2.2 Response-dependent question content algorithm.

1. Respondents are directed to the online survey.

2. The respondent enters covariates

3. R Shiny Server produces questions according to the respondent's covariates

4. The respondent receives the covariate-dependent questions, and answers them.

5. An outcome file is stored locally, or uploaded to an external repository for analysis.

6. If applicable: the respondent receives a random hash code to receive compensation.

An implementation of this algorithm, applied to the study of the interest group endorsement heuristic, is found here: https://github.com/aaronrkaufman/shinySurveys/tree/master/rollcall_lookup.

### 3.2.3 Programmatic question selection algorithm.

1. Respondents are directed to the online survey.

2. Respondents receive and answer the first question.

3. Respondents receive a second question conditional on their first answer, and answer it.

4. Respondents receive a third question conditional on their first two answers, and answer it.

5. . . .

6. Respondents receive the last question conditional on their first N answers, and answer it.

7. An outcome file is stored locally, or uploaded to an external repository for analysis.

8. If applicable: the respondent receives a random hash code to receive compensation.

An implementation of this algorithm, applied to latent trait batteries, is found here: https://github.com/aaronrkaufman/shinySurveys/tree/master/info_battery.

A second implementation, applied to the study of legislative district compactness, is found here: https://github.com/aaronrkaufman/shinySurveys/tree/master/compactness.

### 3.2.4 Sequential pair-randomization algorithm.

1. Respondents are directed to the online survey.

2. Respondents answer pre-treatment demographic variables, and are assigned into random pre-treatment covariates.

3. The latest `blocktools` object is loaded, or downloaded from an external source.

4. The `blocktools` object is updated based on the respondent's pre-treatment covariates, the treatment category is assigned, and the `blocktools` object is overwritten locally, or re-uploaded to the external source.

5. The respondent receives the treatment condition and responds to the outcome question.

6. An outcome file is stored locally, or uploaded to an external repository for analysis.

7. If applicable: the respondent receives a random hash code to receive compensation.

An implementation of this algorithm, applied to a replication of Tomz (2013) studying audience costs, is found here: https://github.com/aaronrkaufman/shinySurveys/tree/master/audience_costs.

## Acknowledgments

The author thanks Matthew Kim for help and inspiration in a previous version of this project, and Ista Zahn for R support and advice. He also thanks Matt Blackwell, Christine Chourat, Tirthankar Dasgupta, Josh Kertzer, Gary King, Mayya Komisarchik, Bob Kubinec, Brian Libgober, Luke Miratrix, Reagan Mozer, Stephen Pettigrew, Jameson Quinn, Erin Rossiter, Don Rubin, Robert Ward, and Madeleine Wolf for helpful conversations, motivating examples, and editorial assistance. Finally, he owes a debt of gratitude to Rick Wilson and two anonymous reviewers for their thoughtful review.

## Author Contributions

**Conceptualization:** Aaron R. Kaufman.

**Investigation:** Aaron R. Kaufman.

**Methodology:** Aaron R. Kaufman.

**Software:** Aaron R. Kaufman.

**Visualization:** Aaron R. Kaufman.

**Writing – original draft:** Aaron R. Kaufman.

**Writing – review & editing:** Aaron R. Kaufman.

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
