## [Decision Letter · Decision Letter 0]

6 Mar 2020

PONE-D-20-00329

Implementing Novel, Flexible, and Powerful Survey Designs in R Shiny

PLOS ONE

Dear Dr. Kaufman,

Thank you for submitting your manuscript to PLOS ONE. After careful consideration, we feel that it has merit but does not fully meet PLOS ONE’s publication criteria as it currently stands. Therefore, we invite you to submit a revised version of the manuscript that addresses the points raised during the review process.

I agree with the reviewers. This manuscript provides a practical tool for the design of surveys. I think that R2’s comments are offered as friendly amendments to the manuscript. You should consider whether you wish to implement them. My sense is that the suggestions offered will strengthen the manuscript.

I look forward to seeing a revision.

We would appreciate receiving your revised manuscript by Apr 20 2020 11:59PM. To enhance the reproducibility of your results, we recommend that if applicable you deposit your laboratory protocols in protocols.io, where a protocol can be assigned its own identifier (DOI) such that it can be cited independently in the future. For instructions see: http://journals.plos.org/plosone/s/submission-guidelines#loc-laboratory-protocols

We look forward to receiving your revised manuscript.

Kind regards,

Rick K. Wilson, Ph.D.

Academic Editor

PLOS ONE

Journal Requirements:

2. Please note that PLOS ONE has specific guidelines on software sharing (http://journals.plos.org/plosone/s/materials-and-software-sharing#loc-sharing-software) for manuscripts whose main purpose is the description of a new software or software package. In this case, new software must conform to the Open Source Definition (https://opensource.org/docs/osd) and be deposited in an open software archive. Please see http://journals.plos.org/plosone/s/materials-and-software-sharing#loc-depositing-software for more information on depositing your software.

4. We note that Figure 1 in your submission contain map images which may be copyrighted. All PLOS content is published under the Creative Commons Attribution License (CC BY 4.0), which means that the manuscript, images, and Supporting Information files will be freely available online, and any third party is permitted to access, download, copy, distribute, and use these materials in any way, even commercially, with proper attribution. For these reasons, we cannot publish previously copyrighted maps or satellite images created using proprietary data, such as Google software (Google Maps, Street View, and Earth). For more information, see our copyright guidelines: http://journals.plos.org/plosone/s/licenses-and-copyright.

Reviewers' comments:

Reviewer's Responses to Questions

**Comments to the Author**

1. Is the manuscript technically sound, and do the data support the conclusions?

Reviewer #1: Yes

Reviewer #2: Yes

2. Has the statistical analysis been performed appropriately and rigorously? 

Reviewer #1: N/A

Reviewer #2: N/A

3. Have the authors made all data underlying the findings in their manuscript fully available?

Reviewer #1: Yes

Reviewer #2: Yes

4. Is the manuscript presented in an intelligible fashion and written in standard English?

Reviewer #1: Yes

Reviewer #2: Yes

5. Review Comments to the Author

Reviewer #1: This paper represents a nice practical solution to a common problem -- the things we can do on surveys in theory given existing statistical tools vastly exceed the things we can put into standard survey software. I suspect these examples will serve as important templates for researchers interested in using advanced survey tools or in designing/testing new tools.

Reviewer #2: Hi Aaron,

Thank you for submitting this manuscript. It was an exciting read and I appreciate the effort you put into the selection of the six important and diverse substantive applications and implementing their corresponding designs. You make a succinct case for R Shiny’s superiority over existing platforms and demonstrate it by example. I recommend some revisions to (1) illustrate your main points upfront particularly in comparison to existing alternatives, (2) better highlight your contribution of open source R Shiny templates, and (3) make clear the broader universe of survey design possibilities to other users with some examples at the end.

On the first note, I would like to see an overview of R Shiny’s benefits upfront, so I recommend placing your list in 2.7 at the start of your article in section 1, perhaps as a subsection. It would be useful to compare this to other alternative(s) which you refer to throughout the article, and explicitly label where R Shiny is stronger/has a feature where the alternative is weak/does not — a table would best show this. Then, in your individual applications, you can refer back to items in the 2.7 list and how your R Shiny tools practically enable them compared to alternatives. In particular, it appears items 2,3,4, and 6 apply in enabling particular applications (out of the six) to be done with ease whereas items 1 and 5 help implementing all of them.

On the second note, I would like for you to better highlight your particular contribution in each of the applications you describe. For instance, in 2.2, you discuss the inefficiency in manually coding up the entire tree-structure; you should refer to the particular algorithm in the Appendix that R shiny offers as a replacement. Similar mentions to the Appendix algorithm section could be done for the other sections. Alternatively, you could visually illustrate these efficiencies rather than just refer to the Appendix algorithms. Related to this latter suggestion, in your 2.7 list, you mention the aesthetic point (6) — it would be relatively easy to have an Appendix section illustrating this aesthetic advantage by comparing some screenshots of your scripts’ UIs to equivalent UI’s that Qualtrics or other tools would force the researchers of some of the six applications into. A major appeal of R Shiny is a visual/graphical one, so there needs to be further proof of this in your paper — the UI screenshots would be sufficient. Minor point, but I would additionally suggest you re-title 2.1, 2.2, 2.4, 2.5 to emphasize the particular point of the application rather than the just the name of the study so that others beyond the scholars specific to those applications can appreciate it. The titling of 2.3 and 2.6 are great in conveying this, so should be left as is.

Finally, I would like a sentence in your concluding remarks suggesting some additional possibilities for survey designs for readers to strengthen your argument about R Shiny’s broad utility. Again, I do not think this requires substantial effort.

In overview, I recommend mostly structural, rather substantive, alterations to your manuscript in order to better highlight the great work you’ve already done. The value of R Shiny, and specifically your particular open source designs, will be made more clear. Thereby, I recommend minor revision. Thanks again for your contribution.

6. PLOS authors have the option to publish the peer review history of their article (what does this mean?). If published, this will include your full peer review and any attached files.

Reviewer #1: No

Reviewer #2: No

---

## [Author Response · Author response to Decision Letter 0]

20 Mar 2020

Dear Professor Wilson and Reviewers, 

I thank you for the opportunity to revise and resubmit my manuscript, and for the excellent and thoughtful feedback provided by the reviewers. I have implemented all suggestions to the best of my ability and I believe it has significantly strengthened the manuscript in framing, scope, and clarity.

On the following pages I respond to each reviewer in turn. Reviewer comments are in italics, followed by my responses. 

Reviewer 1:

“This paper represents a nice practical solution to a common problem -- the things we can do on surveys in theory given existing statistical tools vastly exceed the things we can put into standard survey software. I suspect these examples will serve as important templates for researchers interested in using advanced survey tools or in designing/testing new tools.” 

I am grateful to Reviewer 1 for his or her kind words! R1’s framing of this paper is itself valuable, and I have added language to the introduction of this manuscript reflecting it. 

Reviewer 2: 

“On the first note, I would like to see an overview of R Shiny’s benefits upfront, so I recommend placing your list in 2.7 at the start of your article in section 1, perhaps as a subsection. It would be useful to compare this to other alternative(s) which you refer to throughout the article, and explicitly label where R Shiny is stronger/has a feature where the alternative is weak/does not — a table would best show this.” 

I thank Reviewer 2 for this valuable suggestion. I have placed in Section 1 a discussion of what I consider the three primary advantages of R Shiny for conducting survey research – Integrating External Data, Adaptive Questions, and Custom Interaction Modes – and though I have retained the list in 2.7, I have clarified that it contains what I consider secondary advantages of R Shiny. To the Reviewer’s point that this information is best viewed as a table, I completely agree; I have produced such a table containing both the primary and secondary advantages of R Shiny as compared to Qualtrics and placed it in the first section of the Supplemental Information. I reference it in both Section 1 and Section 2.7 as well.

“Then, in your individual applications, you can refer back to items in the 2.7 list and how your R Shiny tools practically enable them compared to alternatives. In particular, it appears items 2,3,4, and 6 apply in enabling particular applications (out of the six) to be done with ease whereas items 1 and 5 help implementing all of them”.

I have implemented this suggestion based on my revisions above: I have identified for each application which one or more of the three primary advantages I list in Section 1 is most important to that application. 

“On the second note, I would like for you to better highlight your particular contribution in each of the applications you describe. For instance, in 2.2, you discuss the inefficiency in manually coding up the entire tree-structure; you should refer to the particular algorithm in the Appendix that R shiny offers as a replacement. Similar mentions to the Appendix algorithm section could be done for the other sections. Alternatively, you could visually illustrate these efficiencies rather than just refer to the Appendix algorithms.” 

I have implemented this suggestion. I now explicitly reference for each application the Supplemental Information algorithm that solves its research design problem. To complement this I have substantially expanded that section of the Supplemental Information to clarify those algorithms, and included links to the GitHub repository subsection that implements each algorithm.

“Related to this latter suggestion, in your 2.7 list, you mention the aesthetic point (6) — it would be relatively easy to have an Appendix section illustrating this aesthetic advantage by comparing some screenshots of your scripts’ UIs to equivalent UI’s that Qualtrics or other tools would force the researchers of some of the six applications into. A major appeal of R Shiny is a visual/graphical one, so there needs to be further proof of this in your paper — the UI screenshots would be sufficient.” 

I have implemented this important suggestion in two ways. First, I include an image of the R Shiny survey from the application in 2.5, illustrative of interaction modes R Shiny enables. Second, I include in the Supplemental Information a series of images comparing implementations of a conjoint design. The R Shiny implementation is easier to read than the Qualtrics implementation on a desktop web browser, but the Qualtrics mobile implementation is extremely difficult to read whereas R Shiny always produces identical versions across mobile and desktop; this latter point I had not previously made, but as many respondents take surveys on their mobile devices, it is increasingly important.

“Minor point, but I would additionally suggest you re-title 2.1, 2.2, 2.4, 2.5 to emphasize the particular point of the application rather than the just the name of the study so that others beyond the scholars specific to those applications can appreciate it. The titling of 2.3 and 2.6 are great in conveying this, so should be left as is.”

I appreciate R2’s kindness here, and have amended the titles of 2.1, 2.2, 2.4, and 2.5 as suggested.

“Finally, I would like a sentence in your concluding remarks suggesting some additional possibilities for survey designs for readers to strengthen your argument about R Shiny’s broad utility. Again, I do not think this requires substantial effort.”

R2’s suggestion here proved particularly valuable. In considering his or her recommendation, I developed three specific lines of research for which I think R Shiny may prove valuable in the future, the third of which I think has particular promise for addressing measurement error in panel surveys. I believe this paragraph greatly improves the concluding section.

---

## [Editor Report · Decision Letter 1]

2 Apr 2020

PONE-D-20-00329R1

Implementing Novel, Flexible, and Powerful Survey Designs in R Shiny

PLOS ONE

Dear Dr. Kaufman,

Thank you for submitting your manuscript to PLOS ONE. After careful consideration, we feel that it has merit but does not fully meet PLOS ONE’s publication criteria as it currently stands. Therefore, we invite you to submit a revised version of the manuscript that addresses the points raised during the review process.

I appreciate the revisions that you made. I think by following R2’s suggestions you have made the manuscript much clearer.

You do need to pay attention to your final revision. For example, section 2.7 has the title “Additional Benefits, Drawback and Considerations.” But, you have deleted the drawbacks section, so the section title no longer applies. I do not think you need to put drawbacks back into the manuscript – what you previously had is obvious and well known to the community. The comparison table in Supporting Information (section 3.1) is valuable. There may be other places where you need to make changes such that the manuscript conforms to PLOS standards.

We would appreciate receiving your revised manuscript by May 17 2020 11:59PM. To enhance the reproducibility of your results, we recommend that if applicable you deposit your laboratory protocols in protocols.io, where a protocol can be assigned its own identifier (DOI) such that it can be cited independently in the future. For instructions see: http://journals.plos.org/plosone/s/submission-guidelines#loc-laboratory-protocols

We look forward to receiving your revised manuscript.

Kind regards,

Rick K. Wilson, Ph.D.

Academic Editor

PLOS ONE

---

## [Author Response · Author response to Decision Letter 1]

6 Apr 2020

In pursuing greater clarity and ease of readership, I rewrote a number of the section titles, subsection titles, and captions to more clearly reflect their contents and purpose. I also simplified language across all sections.

Secondly, I have amended the manuscript to adhere as closely as I can glean to PLOS guidelines, to whit: I have removed all footnotes, including them in-text where reasonable or excluding them where unnecessary; I have moved all figures and tables to immediately below the first paragraph where they are mentioned; I have converted all citations to the appropriate format; and I clarified a number of abbreviations.

---

## [Editor Report · Decision Letter 2]

15 Apr 2020

Implementing Novel, Flexible, and Powerful Survey Designs in R Shiny

PONE-D-20-00329R2

Dear Dr. Kaufman,

We are pleased to inform you that your manuscript has been judged scientifically suitable for publication and will be formally accepted for publication once it complies with all outstanding technical requirements.

With kind regards,

Rick K. Wilson, Ph.D.

Academic Editor

PLOS ONE

Additional Editor Comments (optional):

Thank you for being so responsive to the reviwers and to my requests. I think the manuscript is set.
---

## [Editor Report · Acceptance letter]

16 Apr 2020

PONE-D-20-00329R2 

Implementing Novel, Flexible, and Powerful Survey Designs in R Shiny 

Dear Dr. Kaufman:

I am pleased to inform you that your manuscript has been deemed suitable for publication in PLOS ONE. Congratulations! Your manuscript is now with our production department. 

With kind regards,

on behalf of

Dr. Rick K. Wilson 

Academic Editor

PLOS ONE